# Susceptibility of Broiler Chickens to Deoxynivalenol Exposure via Artificial or Natural Dietary Contamination

**DOI:** 10.3390/ani11040989

**Published:** 2021-04-01

**Authors:** Regiane R. Santos, Marjolein A. M. Oosterveer-van der Doelen, Monique H. G. Tersteeg-Zijderveld, Francesc Molist, Miklós Mézes, Ronette Gehring

**Affiliations:** 1Schothorst Feed Research, 8200 AM Lelystad, The Netherlands; FMolist@schothorst.nl; 2Veterinary Pharmacotherapy and Pharmacy, Department of Population Health Sciences, Faculty of Veterinary Medicine, Utrecht University, Yalelaan 104-106, 3584 CM Utrecht, The Netherlands; M.A.M.vanderDoelen@uu.nl (M.A.M.O.-v.d.D.); r.gehring@uu.nl (R.G.); 3Institute for Risk Assessment Sciences, Faculty of Veterinary Medicine, Utrecht University, Yalelaan 2, 3584 CM Utrecht, The Netherlands; M.H.G.Tersteeg@uu.nl; 4Faculty of Agricultural and Environmental Sciences, Szent István University, 2100 Gödöllõ, Hungary; Mezes.Miklos@szie.hu

**Keywords:** mycotoxins, fusarium, poultry, nutrient transporter, intestinal morphology

## Abstract

**Simple Summary:**

This study evaluated the effect of diets artificially or naturally contaminated with 4000 μg/kg deoxyvalenol (DON) on the intestinal integrity and nutrient absorption of broiler chickens. Young broiler chickens (14 days old) were more sensitive to DON than older birds (28 days old), and negative impacts were observed when diets were naturally contaminated with DON. Aside from the decrease in the villus height of the jejunum in young broilers, their capacity to absorb peptides was decreased, as shown by the down-regulation of a peptide transporter. However, this effect was compensated in older broilers by an increase in the expression of carbohydrate transporter.

**Abstract:**

Multi-mycotoxin contamination of poultry diets is a recurrent problem, even if the mycotoxins levels are below EU recommendations. Deoxynivalenol (DON) is one of the main studied mycotoxins due to its risks to animal production and health. When evaluating the effects of DON, one must consider that under practical conditions diets will not be contaminated solely with this mycotoxin. In the present study, broiler chickens were fed diets with negligible mycotoxin levels or with naturally or artificially contaminated diets containing approximately 4000 μg/kg DON. Birds were sampled at D14 and D28. Naturally-contaminated diets caused the most harm to the birds, especially the young ones, which presented decreased jejunal villus height and increased lesions, down-regulation of a peptide transporter. At D28 broiler chickens seemed to have adapted to the dietary conditions, when no differences were observed in villus morphometry, together with up-regulation of a carbohydrate transporter. However, intestinal lesions remained present in these older birds.

## 1. Introduction

Feed quality is affected mainly by its ingredients and their processing [1]. The quality of a feed ingredient is not limited to its nutritional composition but will also depend on the presence of contaminants. Mycotoxins are natural and ubiquitous contaminants produced by diverse filamentous fungal species. These toxins are produced already in the field or during grain storage. Deoxynivalenol (DON) is produced in the field by *Fusarium* species and represents one of the most hazardous mycotoxins for farm animals and is often identified during monitoring [2]. The presence of DON in feed will result in suboptimal animal performance [3,4] and, can alter clinical biochemical serum parameters [5] or culminate with secondary disease outbreaks [6,7].

Although pigs are the species most sensitive to DON, this toxin can also impair poultry performance [3,4], hepatic and intestinal function [4,8], and gizzard, thymus and intestine development [5], induce necrotic enteritis [6], and predispose to *Campylobacter jejuni* multiplication in the intestine of broiler chickens [7]. These effects are observed depending on the DON levels in the final diet as well as on the birds’ health status. To assess the production and health risks caused by this mycotoxin, studies have been performed with DON levels usually ranging from 1600 [3] to 15,000 μg/kg [5]. Toxic effects on tissues and organs are usually assessed by testing DON levels far above those recommended by EU guidelines, i.e., 5000 μg/kg [9], and the diet or one of the feedstuffs used in the feed is artificially contaminated with the test mycotoxin at the desired level. In practice, the commonly observed DON levels in poultry diet are below 5000 μg/kg. Aside from this, grains are often contaminated with several mycotoxins [10]. As a result, a diet prepared with different feed ingredients is very likely to be contaminated with a variety of mycotoxins that may interact with each other [11]. Multi-contamination of diets encompasses not only DON and its derivatives (3- and 15-acetyl DON and DON-3-glucoside), but also other parental mycotoxins, such as zearalenone (ZEN) and fumonisins (FUMs), and other neglected ones, e.g., alternariol, beauvericin, and enniatins [3,4,12].

Negative impacts on bird performance can be observed after chronic exposure to a mixture of mycotoxins even when they are all below recommended levels [3,4,13]. There is a strong correlation (r^2^ = 0.85) between the presence of DON in feed and the feed efficiency of broiler chickens [12]. Such a production decrease indicates that instead of producing meat the birds are using their energy for tissue recovery due to mycotoxin injuries or are suffering secondary infections caused by chronic mycotoxicosis. An in vitro study showed that DON inhibits nutrient absorption in human intestinal cells [14], and such an effect may also occur in other animal species fed contaminated diets. The presence of DON in the diet of broiler chickens resulted in gut leakage [8] and paracellular transport of pathogenic bacteria [7] when DON dietary levels were 7500 and 5000 μg/kg, respectively. The study by Osselaere et al. [8] was performed with a naturally-contaminated diet containing DON, its acetylated form 3-acetyl-DON and fumonisins B1, B2, and B3, whereas the study by Ruhnau et al. [7] reported contamination of one of the feedstuffs (wheat) with DON only.

To evaluate the effects of the source of DON contamination in the present study, diets were prepared with different batches of corn, resulting in one diet containing circa 200 μg/kg DON and another with circa 4000 μg/kg DON. Additionally, the diet containing 200 μg/kg DON was artificially contaminated with DON produced by in vitro culture of *Fusarium* in corn, reaching a final concentration of circa 4000 μg/kg DON. Although corn was the main feed contaminant, the combination of all the other feedstuffs to prepare the feed resulted in a multi-contaminated final diet. It was hypothesized that bird age and DON source may affect the intestinal integrity (gut leakage, inflammation, morphometry, lesions, and nutrient transport) of broiler chickens. Therefore, serum markers of gut leakage such as fluorescein isothiocyanate-labelled dextran (FITC-d), D-lactate, and tight-junction-associated protein 1 (TJAP1) were measured. Markers of gut inflammation, such as ovotransferrin and citrulline, were also measured in the serum of broiler chickens. Intestinal morphometry and lesion scores were assessed by histological analysis, and the gene expression of markers for tight-junction proteins, villus integrity and nutrient transport were evaluated by qRT-PCR.

## 2. Materials and Methods

### 2.1. Animals and Housing

One-day-old male Ross 308 broilers were delivered to Schothorst Feed Research (SFR) after hatching and allotted in cages (20 birds per cage) with wood shavings as bedding material, in the broiler facilities of SFR, Lelystad, the Netherlands. Each cage had a surface area of 2.2 m^2^ and contained one feeder and two or three drinking nipples. Two cages per treatment was used and the bird was the experimental unit. A standard broiler temperature schedule was applied. Experimental diets were fed during the experimental period. The ambient temperature was gradually decreased from 34 °C at arrival to 22 °C at 28 days of age. Room temperature and relative humidity were recorded daily. The light was continuously on for the first 24 h to give birds the opportunity to readily find feed and water. After that, the light schedule was 2D:22L during one day, and then changed to 4D:10L:2D:8L during the remaining experimental period, complying with the EU legislation of a minimum of six hours of darkness with at least one period of four hours uninterrupted darkness. Broiler chickens were weighed per pen on days 0 (arrival), 14, and 28.

### 2.2. Test Feedstuffs and Cultured DON

The feedstuffs used in this study were tested for mycotoxins levels to allow the production of diets with marginal and high levels of DON. For this, two corn batches were selected. For the artificial contamination of diets, DON was produced by the *Fusarium graminearum* (NRRL 5883) strain on corn substrate according to Fodor et al. [15]. The cultured DON (16.3 mg DON/g corn substrate) was added in the diet produced with marginally contaminated corn at an appropriate amount of the fungal culture to obtain the expected final DON content.

### 2.3. Diets and Experimental Design

The diets were prepared as mash with naturally contaminated feedstuffs containing marginal or high DON levels. As a DON source, a batch of contaminated corn was used in the present study. The other main feedstuffs (soybean meal and wheat) were present at the same inclusion levels in all diets and had negligible mycotoxin contamination. The experiment comprised in total three dietary treatments and 40 replicates per treatment, where each chicken was a replicate. The birds were fed a corn-based diet with negligible mycotoxin contamination without (control) or with ~4000 μg/kg DON produced via *Fusarium* culture (artificial contamination), or a corn-based diet naturally contaminated with ~4000 μg/kg DON (natural contamination). The experiment had a starter phase from D0-14 and a grower phase from D14-28, and 20 birds per treatment were euthanized at these two time points. The experimental diets were provided for ad libitum intake during the starter and grower phase, respectively. Treatments were randomly allocated to the broiler chickens, which were monitored daily for abnormalities, such as abnormal behavior, clinical signs of illness, and mortality throughout the experiment. Anticipating the potentially increasing impact of mycotoxins when ionophore coccidiostats are banned from feed, the experimental diets were formulated without coccidiostats. Dietary composition is given in Appendix A. Before starting the experiment, all diets were analyzed in an independent and accredited (BELAC 057-TEST/ISO17025) laboratory (Primoris Holding, Ghent, Belgium) via liquid chromatography with tandem mass spectrometry (LC-MS/MS). Mycotoxins levels in the experimental diets are summarized in Table 1 and Table 2. The limit of detection (LOD) were: 1.0 μg/kg for Aflatoxin B1, B2, G1, and G2, ochratoxin A, and sterigmatocystin; 2.0 μg/kg for alternariol, alternariol ME, and cytochalasine; 5.0 μg/kg for beauvericin, diacetoxyscirpenol, enniatin A, A1, B, and B1, and roquefortin C; 10.0 μg/kg for T2 and HT2 toxin sum and citrinin; 15.0 μg/kg for zearalenone; 20.0 μg/kg for DON, 3 + 15 Ac-DON, DON-3-glucoside, and fumonisins B1 + B2; and 50.0 μg/kg for moniliformin and nivalenol.

### 2.4. Gut Leakage Evaluation

To detect gut leakage, a protocol described by Vicuna et al. [16] was applied. In brief, broilers received an oral gavage with FITC-d (MW 3000–5000; Sigma Aldrich Co., St. Louis, MO, USA) diluted in milli-Q water, (2.2 mg/mL; 1 mL/bird) 150 min before euthanasia, after which blood and intestinal samples (jejunum and ileum) were collected. Serum derived from blood (~10 mL collected per bird), was harvested by centrifugation (15 min at 1500× *g*) and stored at −20 °C until analysis. The birds were not fed between inoculation and euthanasia to avoid feed consumption interference.

### 2.5. Measurements

#### 2.5.1. Histological Analysis of Jejunum and Ileum

Jejunum and ileum samples from each bird were submitted to morphometric and morphological analyses as previously described [17]. Histological slides (periodic acid-Schiff (PAS)-hematoxylin staining) from the jejunum and ileum from each bird were scanned by the NanoZoomer-XR (Hamamatsu Photonics KK, Hamamatsu, Japan). The scanned slides were viewed through the viewer software (NDP.view2; Hamamatsu), and analyzed using the analysis software (NDP.analyze; Hamamatsu). Villus height (VH), crypt depth (CD) and villus area (µm^2^) from each individual bird were measured (10 villi per intestinal segment), and the VH:CD ratio was calculated. The number of goblet cells and goblet cells density per villus were also quantified by evaluating the same 10 villi per segment used for the morphometric analysis. For this, scans of Alcian blue stained sections were prepared. Only intact villi were measured. To evaluate the degree of mucosal damage, the Chiu/Park scale was applied [17]. The mucosa was classified from normal if presenting an intact structure with no visible damage (degree 0) to severe damaged (degree 6).

#### 2.5.2. Serum Analysis

For detection of FITC-d levels in serum, fluorescence levels were measured at an excitation wavelength of 485 nm and emission wavelength of 528 nm. Citrulline (ng/mL), ovotransferrin (ng/mL), and tight junction associated protein-1 (ng/mL) were measured using assay kits from MyBiosource Inc. (San Diego, CA, USA), coded as MBS4191177, MBS2610621, and MBS9915242, respectively. Absorbance was measured at a wavelength of 450 nm.

#### 2.5.3. mRNA Expression in Jejunum and Ileum

Segments from jejunum and ileum (10 birds per treatment) were rinsed in cold sterile phosphate buffer saline solution and immediately transferred to RNALater (Qiagen, Hilden, Germany) according to the manufacturer protocol, and stored at −80 °C until processing. RNA from samples of jejunum and ileum was isolated using the SV Total RNA Isolation System (Promega, Madison, WI, USA) according to the manufacturer’s instructions, and total RNA was quantified by spectrophotometer (Nanodrop ND-1000, Thermo Scientific, Wilmington, DE, USA). Subsequently, 1 µg of extracted total RNA was reverse transcribed with the iScriptTM cDNA Synthesis kit (BIO-RAD, Hercules, CA, USA). The cDNA was diluted to a final concentration of 30 ng/µL. Primers, as presented in Table 3, were commercially produced (Eurogentec, Maastricht, the Netherlands). The primers used were selected based on specificity and efficiency by qPCR analysis of dilution series of pooled cDNA at a temperature gradient (55–65 °C) for primer annealing and subsequent melting curve analysis. The reaction mixture for the qPCR containing 10 μL of the diluted cDNA was mixed with 15 μL iQSYBR Green Supermix (Bio Rad Laboratories Inc., Hercules, CA, USA), forward and reverse primers (final concentration of 0.4 pmol/μL for each primer) and sterile water according to the manufacturer’s instructions. qPCR was performed using the MyIQ single-color real-time PCR detection system (Bio-Rad) and MyiQ System Software Version 1.0.410 (Bio Rad Laboratories Inc., Hercules, CA, USA). Data were analyzed using the efficiency-corrected DeltaDelta-Ct method [18]. The fold-change values of the genes of interest were normalized using the geometric mean of the fold-change values of two housekeeping genes. The fold-change values of the target genes were normalized using two housekeeping genes: hypoxanthine-guanine phosphoribosyl transferase (HPRT) and β-actin (ACTB). The mRNA expression of different markers in the jejunum and ileum were measured: mucus production (MUC2), the protein villin, the actin binding protein found in microvilli, which compose brush border (VIL1), inflammation (INFg), tight junction proteins (occludin and ZO-1), and the nutrient transporters PEPT1, GLUT1, and SGLT.

### 2.6. Statistical Analysis

Statistical analysis was carried out with GenStat^®^ for Windows (20th edition; VSN International, Hemel Hempstead, UK). All parameters were analyzed with ANOVA with Fisher’s least significant difference (LSD) post-hoc test to compare treatment means. Values with *p* ≤ 0.05 were considered statistically significant.

## 3. Results

No significant differences in body weight (BW) at D14 and D28 were observed between the challenge and control groups within each feeding period. The average BW of the 14-day-old birds were 394, 393, and 393 g in those fed the control, artificially contaminated and naturally contaminated diets, respectively. The average BWs of the 28-day-old birds were 1249, 1248, and 1184 g in those fed the control, artificially-contaminated, and naturally-contaminated diets, respectively.

### 3.1. Intestinal Morphometry and Morphological Scoring

At D14, a significant decrease in the villus height and increased morphological damage were observed in the jejunum of broilers fed diets contaminated with DON, regardless of its origin. For instance, birds fed the control diet did not present morphologic damage (degree 0), or it was restricted to the villus tips (degree 1), whereas birds fed the DON contaminated diets presented several degrees of damage, mostly characterized by denudation of the villus (degree 3) or denuded villi with dilated capillaries (degree 4).

Additionally, the number of goblet cells in the jejunum of 14-day-old broiler chickens was significantly decreased in birds fed a diet naturally contaminated with DON. No morphometric or morphological alterations were observed in the ileum (Table 4).

At D28, the density of goblet cells in the jejunum of broiler chickens fed the naturally-contaminated diet was significantly increased. Also, 28-day-old broiler chickens exposed to diet naturally contaminated with DON presented significantly greater morphological damage to the jejunum, which was characterized by an increased number of villi with damage in the villus tip (degree 1) and some villi presented an extension of the sub-epithelial space (degree 2). A significant increase in the intestinal damage score was also observed in the ileum, regardless of the DON source in the diet; however, the scores were very low in this intestinal section (Table 5).

Figure 1 illustrates examples of jejunum and ileum sections from 14- and 28-days-old broilers fed the experimental diets.

### 3.2. Serum Analysis

None of the evaluated blood parameters were affected in 14-day-old broilers, regardless of the diet. At D28, broiler chickens fed the diet supplemented with cultured DON presented the significantly highest levels of serum TJAP1. No other significant changes were observed (Figure 2).

### 3.3. mRNA Expression in the Jejunum and Ileum

The effects of experimental diets on the expression of markers of gut integrity and function in the jejunum and ileum are shown in Figure 3. In the jejunum, the expression of VIL1 and PEPT1 was significantly decreased in 14-day-old birds fed the diet naturally contaminated with DON compared to the controls. No changes in the expression of the selected markers were observed in the ileum of these birds.

When assessing the jejunum of 28-day-old broiler chickens, a significant decrease in the expression of INFg was observed when the birds were fed the diets containing DON, regardless of its source. On the other hand, the expression of INFg was significantly increased in the ileum of birds fed the diet contaminated with cultured DON. The expression of GLUT1 was significantly increased in the jejunum and ileum of birds fed the naturally-contaminated diet (Figure 3).

### 3.4. Macroscopic Findings

During sampling of intestinal tissues some remarkable alterations in the liver, kidneys and heart of broiler chickens fed the naturally contaminated diets were observed. At D14, seven of the 20 broilers fed the naturally-contaminated diet presented liquid accumulation in the abdomen, cysts in the liver, hydropericardium and enlarged kidneys (Figure 4), whereas at D28 one broiler presented a bleeding area in the liver, but no other alteration was observed. The liver presenting the bleeding area was prepared for histological analysis, and an acute hemorrhage was detected in the subcapsular region with no major primary pathology (Appendix A). No signs of discomfort among the birds were observed.

## 4. Discussion

In the present study we evaluated the effect of two sources of DON at a dietary concentration of 4000 μg/kg on the intestinal integrity of broiler chickens at 14 and 28 days of age.

Regardless of the DON source, the jejunum of 14-day-old broiler chickens fed diets containing this mycotoxin at levels close to 4000 μg/kg presented the shortest villi. This effect was no longer observed in 28-day-old broilers. Immediately after hatching, intestinal maturation starts and is characterized by morphological and functional changes that take at least four days before the maturation of the jejunal crypt and villi [23]. Intestinal crypts tend to reach a proliferation plateau up to 14 days of age [24]. Deoxynivalenol impairs cell proliferation and causes cell apoptosis [25]. In the jejunum of 14-day-old broilers fed the naturally contaminated diet, the expression of VIL1 was significantly decreased when compared to the controls. Villin is the first protein identified from the microvillus core and is concentrated in the apex of intestinal cells during the morphogenesis of the crypt villus axis and may be related to the renewal of the villus apex [26]. The 14-day-old birds fed diets containing DON, regardless of the source, presented a decreased villus length, which was numerically the lowest in the jejunum from birds fed a diet naturally contaminated with DON. Probably the early exposure to DON retarded cell proliferation and migration to the villi resulting in short jejunal villi, whereas older birds were able to adapt to this condition. This adaptation, however, was limited to villus height because lesions caused by DON were maintained after 28 days of dietary exposure. The jejunum from birds fed the diet artificially contaminated with DON presented an increase in the number of villi with damage in the tip (degree 1), whereas those fed the naturally-contaminated diet presented an increased number of degree 1 damage and sometimes an extension of the sub-epithelial space (degree 2). A similar time-dependent adaptation is observed when evaluating the oxidative stress caused by DON in young broiler chickens [27].

Ileum morphometry was not affected by the diets, irrespective of the age of the birds, and the jejunum was more sensitive than the ileum when exposed to DON, as previously reported [4,5]. This can be explained by the fact that the majority of ingested DON is already absorbed in the proximal parts (duodenum and jejunum) of the small intestine [28]. A decrease in villus height may represent a decrease in the intestinal capacity to absorb nutrients [29], but the uptake of nutrients will also depend on substrate levels, the availability of nutrient transporters, and the turnover rate of enterocytes [30]. To understand this, the expression of the nutrient transporters PEPT1, GLUT1, and SGLT was evaluated in the jejunum and ileum of the broilers. The expression of PEPT1 was significantly decreased in the jejunum of 14-day-old birds fed the diet naturally contaminated with DON compared to the controls. The expression of PEPT1 can be regulated by diet intake, development, hormones, pharmacological agents and pathological conditions [31], and this peptide transporter is up-regulated in situations of feed restriction to increase protein absorption [32], which was not the case in the present study. The down-regulation of PEPT1 in the jejunum indicated that the diet naturally contaminated with DON interfered with peptide transport. A similar effect has already been shown in broiler chickens fed a diet containing DON [33], and this can be explained by a decrease in protein synthesis caused by this mycotoxin [14]. Interestingly, at D28 PEPT1 was no longer down-regulated, but GLUT1 was up-regulated in the jejunum and ileum of birds fed the naturally contaminated diet. The absorption of carbohydrates is mediated by the family of glucose transporters (GLUTs) and sodium-glucose co-transporters (SGLTs). GLUT1 has been identified in all cell types and is responsible for basal levels of glucose uptake, while SGLTs are located on the apical side of intestinal mucosa and have high affinity for sodium-dependent glucose co-transporters [34]. The GLUT1 expression is increased by a decrease in circulating glucose levels [35]. A decrease in villus height induced by dietary DON results in low absorption of glucose in the small intestine of broiler chickens [5]. In the present study, it seems that the increased expression of GLUT1 in both jejunum and ileum occurs as a compensatory process to enhance glucose uptake in the 28-day-old birds. The main difference between the diets containing cultured and natural DON was related to the ~10-fold higher concentration of DON-3-G and 5-fold higher zearalenone (ZEN) concentration in the naturally contaminated diet when compared to that containing cultured DON. Both DON-3-G and ZEN are not a concern for broiler chickens. The derivative DON-3-G is not hydrolyzed in the poultry intestine and cannot bind the ribosome peptidyl transferase center [36]. Although ZEN and its metabolites can be detected in the blood, liver, intestinal content and excreta of poultry, these animals are very tolerant to ZEN [37]. However, it is not known if and how these mycotoxins can favor the toxicity of DON. Piglets exposed to ZEN and its derivatives via lactation present increased DON levels in blood [38]. However, observations in pigs cannot be directly translated to poultry.

At D14, only birds fed naturally contaminated DON presented a significant decrease in jejunal goblet cells, but this decrease was not accompanied by changes in the goblet cell density, indicating that the decreased number might be a result of the decrease in villus length. At D28, however, the density of goblet cells in the jejunum of broilers fed the naturally contaminated diet was significantly increased. It was previously demonstrated that *Salmonella*
*enterica* serovar. Typhimurium infection increases the density of goblet cells in the intestine from broiler chickens [39]. An increase in the number of goblet cells may also represent a protective effect of mucus production when birds are experiencing challenging conditions [40,41]. However, changes in goblet cells density were not accompanied by modulation of MUC-2 expression. Serum levels of citrulline and ovotransferrin were not affected by the dietary treatment. On the other hand, IFNg expression was down-regulated in the jejunum of 28-day-old broiler chickens fed DON, regardless of its source, and up-regulated in the ileum of those 28-day-old birds fed the naturally contaminated diet. Citrulline is mostly produced by enterocytes in the small intestine, especially the duodenum, and is related to enterocyte mass independent of the nutritional and inflammatory status of the animal [42] and was not able to be used as a marker in the present study. Serum ovotransferrin is an acute-phase protein used as a marker of inflammatory and microbial stress in chickens [43]. An increase in the level of this marker is expected during an acute inflammatory process, which was not the case in the present trial. The modulation of IFNg expression dependent on the tissue and DON source confirms that the intestinal sections respond differently to DON exposure, and that the interaction of DON with other mycotoxins will interfere in the expression of this pro-inflammatory cytokine. A decrease in the expression of IFNg was previously demonstrated by Ghareeb et al. [44], when broiler chickens were submitted to a high (10,000 μg/kg feed) chronic exposure to DON. These authors suggested that dietary DON can modify the gene expression of cytokines, impairing the immune system and decreasing disease resistance in poultry. It was remarkable, however, that when broiler chickens were fed cultured DON for 28 days, the ileal expression of IFNg was up-regulated. Up-regulation of IFNg in the cecal tonsils of chickens fed *Furarium* mycotoxin challenged with coccidia has previously been reported [45]. In the present study no coccidiostat was added to the diets, but we did not evaluate the chickens for coccidiosis or related lesions in the ileum. The reason why only the artificially-contaminated diet resulted in IFNg up-regulation remains unknown.

Serum markers related to gut leakage were also evaluated in the present study, but no differences were observed on the expression of occludin or ZO-1 in the jejunum and ileum sections. This may indicate an absence of impact of DON on gut leakage or that these intestinal sections did not respond to the damage caused by DON. The serum levels of TJAP1 were significantly increased in the serum of 28-day-old broiler chickens fed the artificially contaminated diet. Unfortunately, the levels of this protein were evaluated at two single time points, and these levels fluctuate greatly due to the dynamic gut barrier loss and recovery process. To measure properly serum markers of gut leakage it may be necessary also to measure the levels of immunoglobulins IgA and IgG, antibodies against zonulin, occluding, and other tight-junction proteins [46]. Probably the same was true for the FITC-d analysis. During sampling, the presence of cysts in the subcapsular region of the liver was observed. Exposure to *Fusarium* mycotoxins such as DON causes oxidative stress in the liver of broiler chickens [47]. However, the other macroscopic findings showed that alterations were not specific to this organ. Abdominal fluid accumulation, hydropericardium and sometimes enlarged kidneys were also observed in the 14-day-old broiler chickens fed the naturally contaminated diet. Young chickens are more susceptible to the trans-epithelium migration of pathogens [22]. Usually, liver and kidney damage are linked to aflatoxin [48] and ochratoxin A [49] exposure. However, neither of these mycotoxins was detected in the diets. The liver cysts were characterized by a fibrous capsule containing serous fluid, and peripheral daughter cysts were sometimes present. All these symptoms are indicative of bacterial translocation to the liver [50,51], but no microbiological culture was performed to confirm this. Genetic factors can be responsible for the abdominal fluid accumulation and hydropericardium, but the findings in the present study were limited to the birds fed the naturally-contaminated diets. Hollander and Kaunitz [51] summarized the factors involved in leaky gut syndrome, demonstrating that not only paracellular transport but also transcellular uptake mechanisms favor the transmucosal transport of bacteria. At D28 only one bird fed the naturally contaminated diet presented liver hemorrhage. No increased mortality or clinical symptoms were observed during the trial, which suggests that the broilers were able to recover from the pathologies caused by the naturally contaminated diet.

## 5. Conclusions

In conclusion, young broiler chickens (14 days old) were more sensitive to dietary contamination with DON than older birds (28 days old), especially if the diet was prepared with naturally contaminated feedstuffs. Exposure to DON resulted in decreased villus length and decreased expression of the peptide transporter PEPT1, which indicates that DON impairs nutrient absorption not only via a decrease in intestinal absorption surface. Furthermore, at D28 no morphometric differences were observed and the expression of carbohydrate transporter GLUT1 was increased as a compensatory mechanism, although intestinal lesions remained an issue. Even though the birds presented liver cysts, abdominal fluid accumulation, and hydropericardium, no microbiological culture or gram histological staining was performed to conclude that bacterial translocation was taking place. The tested concentration of DON was below EU recommendations and this should be borne in mind when coccidiostat-free diets are prepared.

## Figures and Tables

**Figure 1 animals-11-00989-f001:**
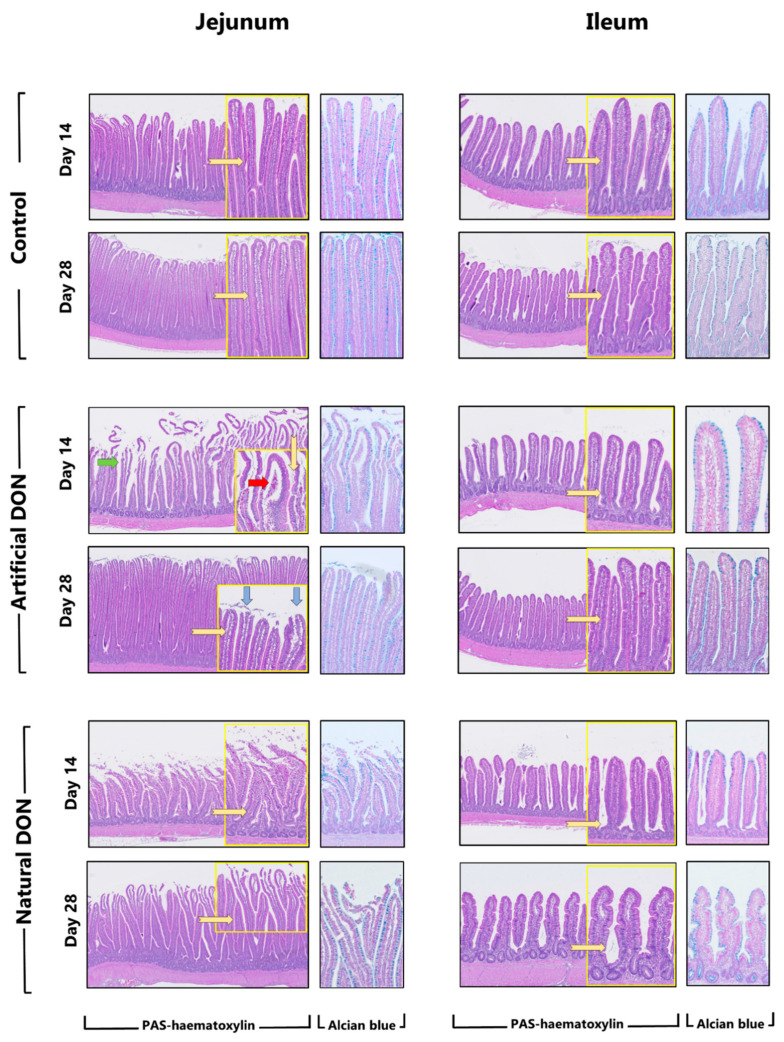
Illustrative images of periodic acid–Schiff (PAS)-hematoxylin- and Alcian blue stained sections of jejunum and ileum from broilers fed the experimental diets; magnification of 200×. Inserts in yellow boxes or with Alcian blue staining are zoomed from the original PAS-hematoxylin staining. The blue arrow indicates denudation of villi tips; the green arrow indicates extensive villus denudation, and the red arrow indicates dilated capillaries.

**Figure 2 animals-11-00989-f002:**
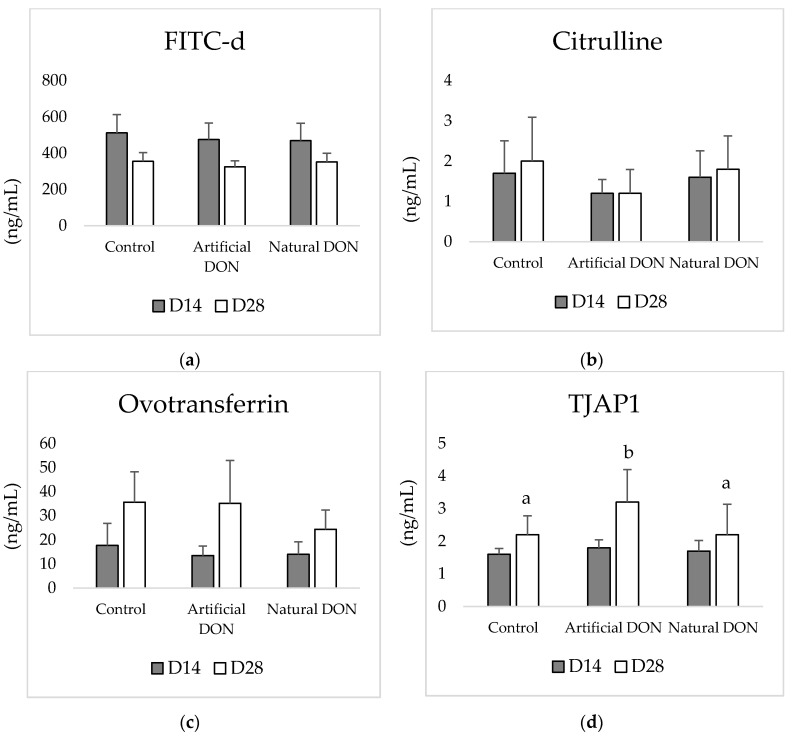
Effect of experimental diets on serum levels of: (**a**) FITC-d; (**b**) citrulline; (**c**) ovotransferrin; (**d**) tight junction associated protein 1 (TJAP1). ^a,b^ Different superscripts indicate significant differences among treatments (*p* ≤ 0.05).

**Figure 3 animals-11-00989-f003:**
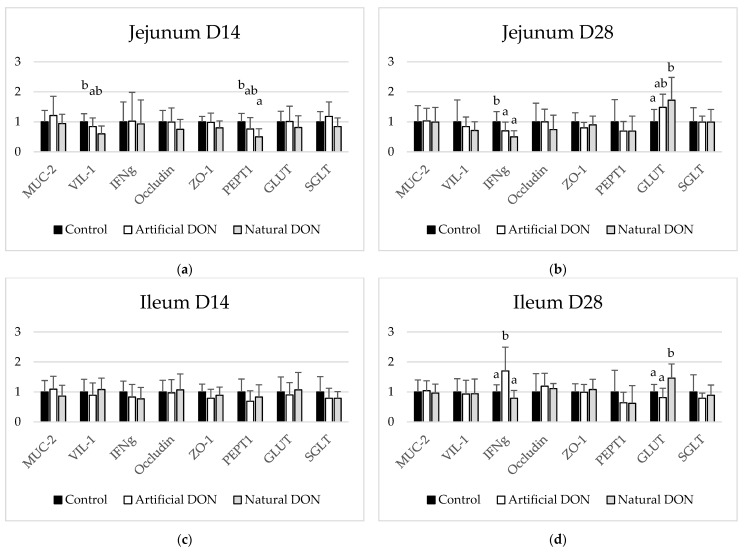
Effect of experimental diets on the mRNA expression of MUC-2, VIL-1, IFNg, occluding, PEPT1, GLUT1, and SGLT in the: (**a**) jejunum of 14-days-old broiler chickens; (**b**) jejunum of 28-days-old broiler chickens; (**c**) ileum of 14-days-old broiler chickens; (**d**) ileum of 28-days-old broiler chickens. ^a,b^ Different superscripts indicate significant differences among treatments (*p* ≤ 0.05).

**Figure 4 animals-11-00989-f004:**
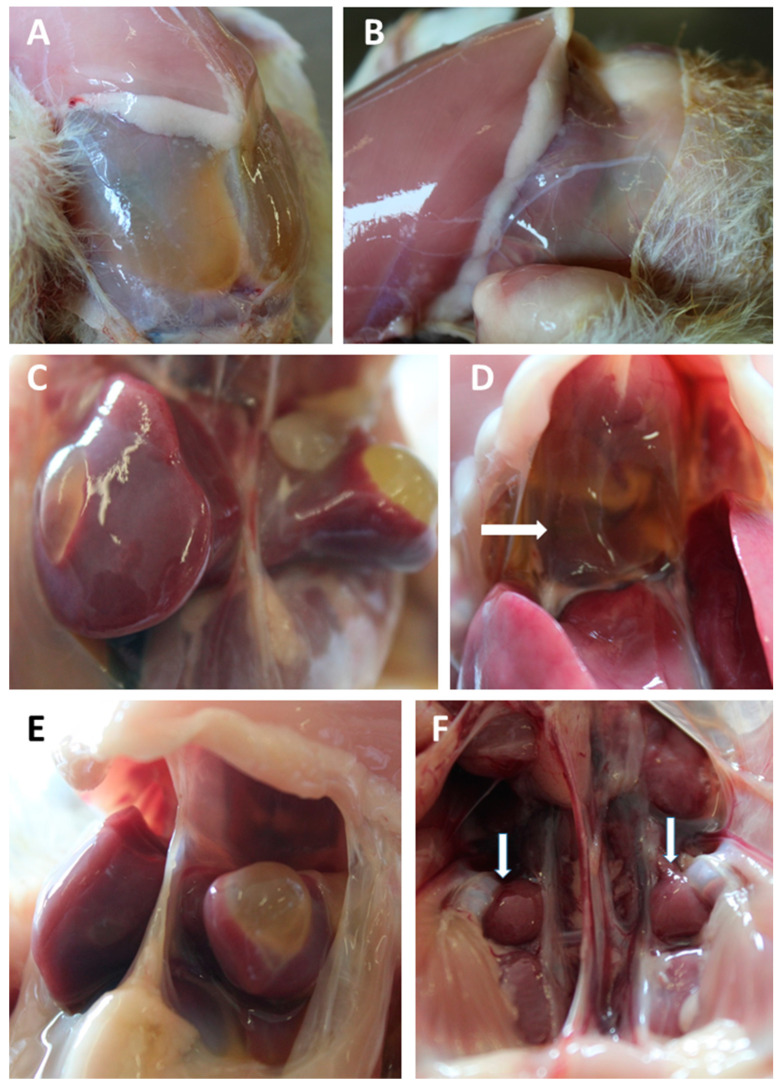
Effect of naturally contaminated diet on 14-days-old broiler chickens, showing: (**A**,**B**) abdominal fluid accumulation; (**C**,**E**) liver cysts; (**D**) hydropericardium (arrow); (**F**) enlarged kidneys (see arrows).

**Table 1 animals-11-00989-t001:** Mycotoxins’ levels (μg/kg) in the experimental treatments in the starter period (D0–14).

Mycotoxins	Control	Artificial DON	Natural DON
DON	233	4040	3810
3 + 15 Ac-DON	-	180	88.1
DON-3-Glucoside	78.3	191	1640
Zearalenone	39.3	105	577
Fumonisins B1 + B2	61.3	103	32.7
Enniatin B	-	6.5	7.8
Beauvericin	8.2	7.7	6.4

Below detection level in all diets: Aflatoxin B1, B2, G1, and G2, nivalenol, ochratoxin A, T2 and HT2 toxin, diacetoxyscirpenol, cytochalasine E, sterigmatocystin, alternariol, alternariol ME, citrinin, roquefortine C, enniatin A, A1, and B1, moniliformin.

**Table 2 animals-11-00989-t002:** Mycotoxins’ levels (μg/kg) in the experimental treatments in the grower period (D14–28).

Mycotoxins	Control	Artificial DON	Natural DON
DON	234	3860	3950
3 + 15 Ac-DON	-	320	59.8
DON-3-Glucoside	75.7	118	895
Zearalenone	31.6	56	727
Aflatoxin B1	1.3	-	-
Aflatoxin B1 + B2 + G1 + G2	1.3	-	-
Fumonisins B1 + B2	56.1	263	-
Enniatin B	-	7.8	-
Beauvericin	5.4	20.9	-

Below detection level in all diets: Nivalenol, ochratoxin A, T2 and HT2 toxin, diacetoxyscirpenol, cytochalasine E, sterigmatocystin, alternariol, alternariol ME, citrinin, roquefortine C, enniatin A, A1, and B1, moniliformin.

**Table 3 animals-11-00989-t003:** Primers used for the expression quantification of housekeeping genes (HKG) and genes of interest (GOI).

Genes	Primer Sequence	Annealing T°	Reference
*HKG*	-	-	-
HPRT	F: GTTGCTGTCTCTACTTAAGCAG R: ATATCCCACACTTCGAGGAG	65	[19]
ACTB	F: ATGTGGATCAGCAAGCAGGAGTA R: TTTATGCGCATTTATGGGTTTTGT	65	[20]
*GOI*	-	-	-
MUC-2	F: ATGCGATGTTAACACAGGACTC R: GTGGAGCACAGCAGACTTTG	61	[21]
VIL-1	F: GGCACCAACGAGTACAACACCA R: TGCAGCCCTTCCCATACCAGA	65	[20]
IFNg	F: CAAGCTCCCGATGAACGAC R: CAATTGCATCTCCTCTGAGAC	64	[20]
Occludin	F: TTCATGATGCCTGCTCTTGTG R: CCTGAGCCTTGGTACATTCTTGT	61	[20]
ZO-1	F: CTTCAGGTGTTTCTCTTCCTCCTC R: CTGTGGTTTCATGGCTGGATC	59	[19]
PEPT-1	F: CCCCTGAGGAGGATCACTTGTT R: CAAAAGAGCAGCAGCAACGA	59	[19]
GLUT1	F: TTGCTGGCTTTGGGTTGTG R: GGAGGTTGAGGGCCAAAGTC	57	[20]
SGLT	F: TGTCTCTCTGGCAAGAACATGTC R: GGGCAAGAGCTTCAGGTATCC	59	[22]

**Table 4 animals-11-00989-t004:** Effect of experimental diets on morphometric and morphological parameters in jejunum and ileum from 14-days-old broiler chickens.

Intestinal Section	Control	Artificial DON	Natural DON	*p*-Value	LSD
*Jejunum*	-	-	-	-	-
Villus height (μm)	873.3 ^b^	713.2 ^a^	656.4 ^a^	0.02	152.84
Crypt depth (μm)	168.5	134.4	144.6	0.09	31.06
VH/CD	5.34	5.66	4.98	0.51	1.18
Villus area (μm^2^)	93.2	81.9	60.8	0.18	34.88
Goblet cells (number)	142.0 ^b^	125.2 ^ab^	93.1 ^a^	0.02	32.45
Goblet cells density ^1^	1.5	1.7	1.6	0.50	0.29
Integrity score ^2^	0.84 ^a^	3.24 ^b^	3.48 ^b^	< 0.001	0.98
*Ileum*	-	-	-	-	-
Villus height (μm)	493.2	450.1	489.4	0.52	84.46
Crypt depth (μm)	134.8	115.4	134.0	0.22	25.14
VH/CD	3.80	4.02	3.88	0.89	0.93
Villus area (μm^2^)	45.7	39.5	44.8	0.24	7.99
Goblet cells (number)	99.5	81.4	99.3	0.13	19.95
Goblet cells density ^1^	2.2	2.2	2.4	0.63	0.05
Integrity score ^2^	0.12	0.14	0.24	0.79	0.37

^a,b^ Different superscripts indicate significant differences among treatments (*p* ≤ 0.05). ^1^ Number of goblet cells per μm^2^ of a villus. ^2^ Integrity score: the higher the score the worse the integrity.

**Table 5 animals-11-00989-t005:** Effect of experimental diets on morphometric and morphological parameters in jejunum and ileum from 28-days-old broiler chickens.

Intestinal Section	Control	Artificial DON	Natural DON	*p*-Value	LSD
*Jejunum*	-	-	-	-	-
Villus height (μm)	1277	1224	1144	0.35	185.49
Crypt depth (μm)	287.6	300.4	259.1	0.45	67.40
VH/CD	4.61	4.46	4.59	0.95	0.96
Villus area (μm^2^)	209.2	213.4	155.0	0.47	107.23
Goblet cells (number)	165.1	156.6	169.5	0.78	37.99
Goblet cells density ^1^	0.80 ^a^	0.91 ^a^	1.20 ^b^	0.02	0.23
Integrity score ^2^	0.38 ^a^	0.62 ^ab^	1.32 ^b^	0.04	0.734
*Ileum*	-	-	-	-	-
Villus height (μm)	682.5	600.0	637.5	0.27	102.23
Crypt depth (μm)	188.7	179.2	169.2	0.63	40.67
VH/CD	3.82	3.55	3.96	0.51	0.724
Villus area (μm^2^)	119.15	90.62	80.85	0.16	40.54
Goblet cells (number)	144.2	121.3	129.6	0.20	25.71
Goblet cells density ^1^	1.40	1.60	1.80	0.22	0.43
Integrity score ^2^	0.01 ^a^	0.06 ^b^	0.04 ^b^	0.03	0.039

^a,b^ Different superscripts indicate significant differences among treatments (*p* ≤ 0.05). ^1^ Number of goblet cells per μm^2^ of a villus. ^2^ Integrity score: the higher the score the worse the integrity.

## Data Availability

Data sharing not applicable.

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
