# Peer review of "Susceptibility of Broiler Chickens to Deoxynivalenol Exposure via Artificial or Natural Dietary Contamination"

_animals, 2021, doi:10.3390/ani11040989_

Round 1

Reviewer 1 Report

Dear Editor,

 The manuscript “Susceptibility of broiler chickens to deoxynivalenol exposure 2 via artificial or natural dietary contamination” presents new information concerning the effects of deoxynivalenol in the intestine of birds. The authors have evaluated two different ages of birds (14 and 28-days) showing a different response for each age. According to the results, the capacity to absorb peptides is decreased in young broilers, while in older this effect was counteracted by an increase in the expression of carbohydrate transporter.

The manuscript is well-written, the aim is clear, however some issues must be improved and clarified.

Introduction

Page 2, line 46-47: The presence of DON in feed will result in suboptimal animal performance and can culminate in clinical symptoms or secondary disease 47 outbreaks.

Please insert a reference for this statement.

03- Material and Methods.

Page 3, lines 97-99

2.1 Animals and housing.  One-day-old male Ross 308 broilers were delivered to SFR after hatching and allotted 97 in cages (20 birds per cage) with wood shavings as bedding material, in the broiler facilities of SFR, Lelystad, the Netherlands.

Please inform the meaning of the acronym SFR.

2.3 Diets and experimental design

It will be interesting to add the limits of detection for the mycotoxins analyzed.

Line 146 - 2.4 Oral gavage

Please change the subtitle to “Gut leakage evaluation” and inform how the serum samples were stored.

Line 160-163 - 2.5.1 Histological analysis of jejunum and ileum.

Why only five villi were analyzed? In general, 30 villi/crypt per segment were evaluated. The authors refer to a previous study (Santos et al, 2015) as a model of histological analysis, but in this study at least 10 villi per animal were evaluated.

Please inform how many villi were used for the evaluation of the number and density of goblet cells. Please clarify how the density of goblet cells was established. Was the number of goblet cells counted using HE-stained slides? Or special staining was prepared?

Line 173- 2.5.3 mRNA expression in jejunum and ileum.

Please inform how the intestine was sampled and stored, and how many birds per treatment were evaluated.

Line 196 – please correct occludin.

 Results

Line 205 – bodyweight. Please inform in material and methods that the animals were weighed and when. Only in 14 and 28 days of the experiment or weekly?

Line 213 and line 224– please describe the main histological findings observed in control and DON groups. The authors refer to damage, but it is important to specify of kind of change that was detected. Considering that the integrity score increased 3.5-fold in the jejunum of animals fed naturally contaminated diet, the description of histological aspects is needed.

Figure 1. In the legend inform the meaning of the acronym TJAP1

Figure 3 legend - (a,b) abdominal water accumulation.

Please correct:  fluid accumulation.

Figure 4 legend - d) inflammation with neutrophils being observed in another insert of the liver hemorrhagic 274 area (see arrows).

The figure shows a focally extensive hemorrhagic area in the liver, so the presence of polymorphonuclear cells is expected as the major type of leukocyte in the blood. The interpretation as an inflammatory infiltrate is not correct. Furthermore, polymorphonuclear cells in broilers are heterophils, not neutrophils.

Figure 4 is unnecessary. My suggestion is to add histological aspects of intestinal changes instead of hemorrhage in the liver.

Figure 4. Insert a measurement object (bar) or inform the magnification for histological sections.

Discussion

Line 279 - Anticipating the potentially increasing impact of mycotoxins when ionophore anti-279 coccidials are banned from feed, the experimental diets were formulated without coccidiostats.

Please change this information to the material and methods section.

Line 285 – Please correct characterised to characterized.

Line 297 – the description of histological changes should also be discussed. Villi height showed no difference in 28-day broilers; however, damage persists. Which type of damage/lesion?

Line 341-342 - the density of goblet cells in the jejunum of broilers fed the naturally contaminated diet was significantly increased, indicating an inflammatory response.

Please insert a reference to this statement (Increase in goblet cells as an indicator of inflammatory response).

Line 382 - These signals are indicative of bacterial translocation. Please insert a reference to this statement

Line 386- Please correct characterised to characterized.

The liver cysts were characterised by a thick wall and floating inclusions….

Please clarify what are floating inclusions and the distribution of cysts (multifocal, focal?)  The concept of cysts is a cystic space delimited by a fibrous capsule containing serous fluid.

Conclusions

Line 403 – 407 Exposure to a naturally contaminated diet also resulted in bacterial translocation in young birds and this effect was not measured by serum markers even when birds presented liver cysts, abdominal liquid accumulation and hydropericardium.

The results do not permit this conclusion. The authors have not performed an evaluation of bacterial translocation through microbiological culture, PCR, or Gram histological staining, so this conclusion is not evidenced by the results. The authors can suggest that DON induces bacterial translocation. It is also important to remember that genetic factors can be responsible for the abdominal liquid and hydropericardium, besides that the liver cysts and hemorrhage can also be the result of traumas or other infectious conditions.

https://doi.org/10.1590/S0100-204X2002000900001 

Author Response

Reviewer 1

The manuscript “Susceptibility of broiler chickens to deoxynivalenol exposure via artificial or natural dietary contamination” presents new information concerning the effects of deoxynivalenol in the intestine of birds. The authors have evaluated two different ages of birds (14 and 28-days) showing a different response for each age. According to the results, the capacity to absorb peptides is decreased in young broilers, while in older this effect was counteracted by an increase in the expression of carbohydrate transporter. The manuscript is well-written, the aim is clear, however some issues must be improved and clarified.

A: We acknowledge the positive comments and the pertinent suggestions/corrections made by the present reviewer. All the requests were performed and are marked with track changes.

Introduction

Page 2, line 46-47: The presence of DON in feed will result in suboptimal animal performance and can culminate in clinical symptoms or secondary disease outbreaks. Please insert a reference for this statement.

A: Clinical symptoms was corrected to clinical biochemical serum parameters. References are now inserted.

03- Material and Methods.

Page 3, lines 97-99

2.1 Animals and housing.  One-day-old male Ross 308 broilers were delivered to SFR after hatching and allotted 97 in cages (20 birds per cage) with wood shavings as bedding material, in the broiler facilities of SFR, Lelystad, the Netherlands. Please inform the meaning of the acronym SFR.

A: Full name is now given: Schothorst Feed Research

2.3 Diets and experimental design

It will be interesting to add the limits of detection for the mycotoxins analysed.

A: All the limits of detection are now given.

Line 146 - 2.4 Oral gavage

Please change the subtitle to “Gut leakage evaluation” and inform how the serum samples were stored.

A: The subtitle was changed as requested and the information on the harvesting and storage of the serum samples is now given.

Line 160-163 - 2.5.1 Histological analysis of jejunum and ileum.

Why only five villi were analyzed? In general, 30 villi/crypt per segment were evaluated. The authors refer to a previous study (Santos et al, 2015) as a model of histological analysis, but in this study at least 10 villi per animal were evaluated.

Please inform how many villi were used for the evaluation of the number and density of goblet cells. Please clarify how the density of goblet cells was established. Was the number of goblet cells counted using HE-stained slides? Or special staining was prepared?

A: Actually we evaluated 10 villi as previously described (Santos et al., 2015). In another recently published study (Santos and van Eerden, 2021) we evaluated five villi per segment.  In the past, we found no differences when comparing data obtained with 5, 10 or 30 villi/crypt per segment (data not published). For the evaluation of the number and density of goblet cells, the same 10 villi were evaluated. Used staining were HE and Alcian Bleu. This complete information is now given.

Line 173- 2.5.3 mRNA expression in jejunum and ileum.

Please inform how the intestine was sampled and stored, and how many birds per treatment were evaluated.

A: All the information is now given with details.

Line 196 – please correct occludin.

A: Corrected.

Results

Line 205 – bodyweight. Please inform in material and methods that the animals were weighed and when. Only in 14 and 28 days of the experiment or weekly?

A: This is now informed in the Material and methods section. The birds were weighed at arrival (D0), D14 and D28 only. Due to the number of used animals, no performance production was evaluated.

Line 213 and line 224– please describe the main histological findings observed in control and DON groups. The authors refer to damage, but it is important to specify of kind of change that was detected. Considering that the integrity score increased 3.5-fold in the jejunum of animals fed naturally contaminated diet, the description of histological aspects is needed.

A: This is now informed with details.

Figure 1. In the legend inform the meaning of the acronym TJAP1

A: This is now informed (now Figure 2).

Figure 3 legend - (a,b) abdominal water accumulation. Please correct:  fluid accumulation.

A: Corrected (now Figure 4).

Figure 4 legend - d) inflammation with neutrophils being observed in another insert of the liver hemorrhagic 274 area (see arrows). The figure shows a focally extensive hemorrhagic area in the liver, so the presence of polymorphonuclear cells is expected as the major type of leukocyte in the blood. The interpretation as an inflammatory infiltrate is not correct. Furthermore, polymorphonuclear cells in broilers are heterophils, not neutrophils.

A: We acknowledge the present referee for this correction. The information is corrected. The Figure 4 is now the Supplementary Figure 1. Another figure (now Figure 1) was added to depict intestinal images.

Figure 4 is unnecessary. My suggestion is to add histological aspects of intestinal changes instead of hemorrhage in the liver.

A: A figure (now Figure 1) with intestinal representative images was included in the revised version. The Figure 4 is now the supplementary Figure 1.

Figure 4. Insert a measurement object (bar) or inform the magnification for histological sections.

A: This is now Supplementary Figure 1. Magnification is now given.

Discussion

Line 279 - Anticipating the potentially increasing impact of mycotoxins when ionophore anti- coccidials are banned from feed, the experimental diets were formulated without coccidiostats. Please change this information to the material and methods section.

A: This sentence was moved to Material and Methods section as requested (subtopic 2.3 Diets and experimental design).

Line 285 – Please correct characterised to characterized.

A: Corrected.

Line 297 – the description of histological changes should also be discussed. Villi height showed no difference in 28-day broilers; however, damage persists. Which type of damage/lesion?

A: Detailed information is now given.

Line 341-342 - the density of goblet cells in the jejunum of broilers fed the naturally contaminated diet was significantly increased, indicating an inflammatory response. Please insert a reference to this statement (Increase in goblet cells as an indicator of inflammatory response).

A: The sentence was not properly written and it was replaced by the following text: “. It was previously demonstrated that Salmonella Typhimurium infection increases the density of goblet cells in the intestine from broiler chickens [38]. An increase in the number of goblet cells may also represent a protective effect of mucus production when birds are experiencing challenging conditions [39,40]. However, changes in goblet cells density were not accompanied by modulation of MUC-2 expression”.

Line 382 - These signals are indicative of bacterial translocation. Please insert a reference to this statement

A: Based on the corrections and comments of the present referee, this sentence was deleted.

Line 386- Please correct characterised to characterized.

A: Corrected.

The liver cysts were characterised by a thick wall and floating inclusions….

Please clarify what are floating inclusions and the distribution of cysts (multifocal, focal?)  The concept of cysts is a cystic space delimited by a fibrous capsule containing serous fluid.

A: Our description of cyst was confusing and the floating inclusion was meant to be serous fluid. Therefore, we kept it as “The liver cysts were characterized by a fibrous capsule containing serous fluid...”

Conclusions

Line 403 – 407 Exposure to a naturally contaminated diet also resulted in bacterial translocation in young birds and this effect was not measured by serum markers even when birds presented liver cysts, abdominal liquid accumulation and hydropericardium. The results do not permit this conclusion. The authors have not performed an evaluation of bacterial translocation through microbiological culture, PCR, or Gram histological staining, so this conclusion is not evidenced by the results. The authors can suggest that DON induces bacterial translocation. It is also important to remember that genetic factors can be responsible for the abdominal liquid and hydropericardium, besides that the liver cysts and hemorrhage can also be the result of traumas or other infectious conditions.

 A: Conclusion section was adapted accordingly, and mention to bacterial translocation in the simple summary and abstract sections was deleted.

Reviewer 2 Report

The manuscript “Susceptibility of broiler chickens to deoxynivalenol exposure via artificial or natural dietary contamination” describes the effects of artificial and natural deoxynivalenol (DON) contaminated diet on intestinal health of broiler chickens. Results show that the intestinal morphological parameters were worse and peptide transporter related gene expression was decreased by DON contaminated diet at D14. While, the morphological parameters were not show defined differences, and glucose transporter gene expression was increased at D28. In addition, bacterial translocation signal was detected, but the levels of gut leakage marker were not changed by DON at D14.

The manuscript is including interesting and important data for poultry industry, while the manuscript is including some issues should be addressed, as below.

Major

[1] Did you check body weight of experimental birds in this study? Authors considered changes of gene expressions of peptide and glucose transporters. The record of body weight is looks important for the relationship with nutrient transporters.

[2] Why the effects of DON on intestinal morphology were stronger in jejunum than ileum?

[3] Authors showed details of bleeding area in the liver in figure 4, but the symptom is rare case in this study (only one bird in total 20 birds). I recommend the data should be moved to supplementary file, because the data is not important essence in this study. I also think that the microscope images of intestinal morphology are more adequately to add in figure than the liver image, because the intestinal morphology showed significantly differences in table 4 and 5.

Minor

L126 “DON (natural conatamination.” Is should be addressed to “DON (natural conatamination).”

Fig. 3 and 4. Letters in figure are upper case, but in legend are lower case. Authors should unify it.

Author Response

Reviewer 2

The manuscript “Susceptibility of broiler chickens to deoxynivalenol exposure via artificial or natural dietary contamination” describes the effects of artificial and natural deoxynivalenol (DON) contaminated diet on intestinal health of broiler chickens. Results show that the intestinal morphological parameters were worse and peptide transporter related gene expression was decreased by DON contaminated diet at D14. While, the morphological parameters were not show defined differences, and glucose transporter gene expression was increased at D28. In addition, bacterial translocation signal was detected, but the levels of gut leakage marker were not changed by DON at D14. The manuscript is including interesting and important data for poultry industry, while the manuscript is including some issues should be addressed, as below.

A: We acknowledge the positive comments and pertinent suggestions/corrections made by the present reviewer. All the requests were performed and are marked with track changes.

Major

[1] Did you check body weight of experimental birds in this study? Authors considered changes of gene expressions of peptide and glucose transporters. The record of body weight is looks important for the relationship with nutrient transporters.

A: Body weight of the birds was assessed at days 0, 14, and 28. This information is now given in the material and methods section. Although the birds fed the naturally contaminated diet were lighter than the birds fed the other diets, this difference was not significant. Moreover, to compare the results of mRNA expression with performance, a much greater number of experimental broiler chickens is needed.

[2] Why the effects of DON on intestinal morphology were stronger in jejunum than ileum?

A: The majority of the ingested DON is already absorbed in the proximal parts of the small intestine. An explanation with a reference is now given.

[3] Authors showed details of bleeding area in the liver in figure 4, but the symptom is rare case in this study (only one bird in total 20 birds). I recommend the data should be moved to supplementary file, because the data is not important essence in this study. I also think that the microscope images of intestinal morphology are more adequately to add in figure than the liver image, because the intestinal morphology showed significantly differences in table 4 and 5.

A: As requested, the original Figure 4 is now the Supplementary Figure 1. And a figure was added (Figure 1) showing intestinal morphology.

Minor

L126 “DON (natural conatamination.” Is should be addressed to “DON (natural conatamination).”

A: Corrected.

Fig. 3 and 4. Letters in figure are upper case, but in legend are lower case. Authors should unify it.

A: Corrected.

Round 2

Reviewer 1 Report

The authors have performed the suggested changes, including new figures showing intestinal changes. The manuscript can be accepted in the present form.